

# Bacteriogenic synthesis of morphologically diverse silver nanoparticles and their assessment for methyl orange dye removal and antimicrobial activity

Bhakti Patel[1],[*], Virendra Kumar Yadav[1],[*], Reema Desai[1], Shreya Patel[1], Abdelfattah Amari[2], Nisha Choudhary[1], Haitham Osman[2], Rajat Patel[1], Deepak Balram[3], Kuang-Yow Lian[3], Dipak Kumar Sahoo[4] and Ashish Patel[1]

[1] Department of Life Sciences, Hemchandracharya North Gujarat University, Patan, Gujarat, India
[2] Department of Chemical Engineering, College of Engineering, King Khalid University, Abha, Saudi Arabia
[3] Department of Electrical Engineering, National Taipei University of Technology, Taipei, Taiwan
[4] Department of Veterinary Clinical Sciences, Iowa State University, Ames, Iowa, United States
* These authors contributed equally to this work.

Corresponding authors
Dipak Kumar Sahoo,
dsahoo@iastate.edu
Ashish Patel, uni.ashish@gmail.com

## ABSTRACT

Nanotechnology and nanoparticles have gained massive attention in the scientific community in recent years due to their valuable properties. Among various AgNPs synthesis methods, microbial approaches offer distinct advantages in terms of cost-effectiveness, biocompatibility, and eco-friendliness. In the present research work, investigators have synthesized three different types of silver nanoparticles (AgNPs), namely AgNPs-K, AgNPs-M, and AgNPs-E, by using *Klebsiella pneumoniae* (MBC34), *Micrococcus luteus* (MBC23), and *Enterobacter aerogenes* (MBX6), respectively. The morphological, chemical, and elemental features of the synthesized AgNPs were analyzed by using UV-Vis spectroscopy (UV-Vis), Fourier transform-infrared spectroscopy (FTIR), X-ray diffraction (XRD), field emission scanning electron microscope (FESEM) and energy-dispersive spectroscopy (EDX). UV-Vis absorbance peaks were obtained at 475, 428, and 503 nm for AgNPs-K, AgNPs-M, and AgNPs-E, respectively. The XRD analysis confirmed the crystalline nature of the synthesized AgNPs, having peaks at $26.2°$, $32.1°$, and $47.2°$. At the same time, the FTIR showed bands at 599, 963, 1,693, 2,299, 2,891, and 3,780 $cm^{-1}$ for all the types of AgNPs indicating the presence of bacterial biomolecules with the developed AgNPs. The size and morphology of the AgNPs varied from 10 nm to several microns and exhibited spherical to porous sheets-like structures. The percentage of Ag varied from 37.8% (wt.%) to 61.6%, *i.e.*, highest in AgNPs-K and lowest in AgNPs-M. Furthermore, the synthesized AgNPs exhibited potential for environmental remediation, with AgNPs-M exhibiting the highest removal efficiency (19.24% at 120 min) for methyl orange dye in simulated wastewater. Further, all three types of AgNPs were evaluated for the removal of methyl orange dye from the simulated wastewater, where the highest dye removal percentage was 19.24% at 120 min by AgNPs-M. Antibacterial potential of the synthesized AgNPs assessment against both Gram-positive (GPB) *Bacillus subtilis* (MBC23), *B. cereus* (MBC24), and Gram-negative bacteria *Enterococcus faecalis* (MBP13) revealed promising results,

with AgNPs-M, exhibiting the largest zone of inhibition (12 mm) against GPB
*B. megaterium*. Such investigation exhibits the potential of the bacteria for the
synthesis of AgNPs with diverse morphology and potential applications in
environmental remediation and antibacterial therapy-based synthesis of AgNPs.

**Subjects** Biochemistry, Microbiology, Soil Science, Environmental Contamination and
Remediation
**Keywords** Silver nanoparticle, Methyl orange, Bioremediation, Antimicrobial, Zone of inhibition

# INTRODUCTION

The rapid industrialization in India as well as in the whole world, has increased the use of
different synthetic dyes. Some of the dyes may cause harm to aquatic life and cause diseases
in living organisms (*Patel et al., 2022*). Dyes are mainly used in textile industries for
coloring fabric, so textile industrial wastewater acts as a significant source of dye effluent
(*Al-Tohamy et al., 2022*; *Deng & Brillas, 2023*; *Kurade et al., 2023*). The prolonged and
continuous mixing of dye-laden water and dye effluents in the freshwater may lead to
water pollution (*Patel et al., 2022*). The consumption of dye-contaminated water may
cause numerous diseases in humans, like skin irritation and skin cancer, in the long term
(*Ekanayake, Udayanga & Manage, 2022*; *Qazi et al., 2022*; *Sagadevan et al., 2022*; *Shan
et al., 2023*). Dyes present in textile effluent can be removed by using various chemical
approaches like precipitation, coagulation (*Wang et al., 2023b*), flocculation, membrane
filtration (nanofiltration, ultrafiltration) (*Wang et al., 2023c*), reverse osmosis, adsorption,
*etc*. (*Robati et al., 2016*). The biological methods involve the utilization of microorganisms
(*Gupta et al., 2022*) for dye remediation, either in the natural sites or in the bioreactor in
the laboratory (*Das & Mishra, 2017*). The biological approach also involves biosorbents for
the remediation of dyes from wastewater (*Cui et al., 2017*). Such processes are economical
if the biosorbents are developed from agricultural waste, *etc*. All of these processes have
certain advantages and disadvantages, but adsorption is a simple, effective, economical
approach where adsorbent can be easily surface functionalized for the removal of specific
pollutants (*Harja, Buema & Bucur, 2022*). The various adsorbents that are commonly used
for the remediation of dyes and other pollutants from contaminated water are alumina,
silica (*Imoisili, Nwanna & Jen, 2022*), zeolites (*Murukutti & Jena, 2022*), coal fly ash,
magnetite, maghemite, zinc oxide (*Fouda et al., 2023*), titanium dioxide (*Yang, Shojaei &
Shojaei, 2022*; *Rathore et al., 2024*), and other complexes (*Cui et al., 2017*). When these
adsorbents are used in their nano form, they become highly effective due to their greater
surface area to volume ratio (SVR) and high surface energies (*Chen et al., 2020*).

Nanotechnology has played a significant role in environmental clean-up, especially the
removal of textile dyes from contaminated water. Among NPs, metallic, metal oxide NPs,
and nanocomposites have been used widely for this application. Among pure metallic NPs,
silver nanoparticles (AgNPs) (*Long et al., 2023*; *Guo et al., 2023*) have gained massive
popularity as they are effective in killing waterborne pathogens due to their antimicrobial
properties (*Sun et al., 2022*; *Puri, Gupta & Mishra, 2021*; *Degefa et al., 2021*; *Tarekegn et al.,
2021*; *Wang et al., 2023a*). Being a heavy metal, Ag coagulates the enzymes and proteins of
the microorganism, thus inhibiting them and ultimately killing them (*Betts, Whitehead & Harris, 2021*). The small size and high SVR of AgNPs enhance their antibacterial effectiveness by allowing them to easily penetrate the cell wall of microorganisms, inhibiting and killing them (*Cui et al., 2013*; *Karunakaran et al., 2017*). The smaller size of AgNPs may facilitate their entry into the microbes and exhibit their antimicrobial effect. AgNPs could be synthesized by all three possible methods, *i.e.*, chemical, physical, and biological (*Bouafia et al., 2021*). The chemical approaches include chemical coprecipitation (*Adibah, Firdianti & Suprapto, 2023*), sol-gel (*Shahjahan et al., 2017*), chemical reduction (*Horne et al., 2023*), polyol (*Wolf et al., 2022*), *etc.* Among physical approaches, the most familiar ones are ball milling (*Lai et al., 2023*), the vapor condensation method (*Simchi et al., 2007*), arc discharge (*El-Khatib et al., 2018*), and laser-ablation techniques (*Rafique et al., 2019*; *Juma et al., 2023*). The chemical approaches involve the utilization of hazardous chemicals, which are not environmentally friendly, while the physical ones utilize expensive instruments, making the synthesis highly expensive. The biological synthesis techniques, *i.e.*, phytonanofabrication (*Choudhary et al., 2023*) and microbial synthesis of AgNPs, are of high significance due to the lower utilization of chemicals and the absence of the requirement of a chemical capping agent or surfactant. The microbial synthesis methods provide natural stabilizing and capping agents, which make them biocompatible (*Naganthran et al., 2022*). Among all the microorganisms, bacterial synthesis is preferred due to their easy handling and short duration for their growth compared to algae and fungi. Bacteria are enriched with several bacterial enzymes and proteins, which play an essential role in the bio-reduction of $Ag^+$ ions into $Ag^0$. These biological molecules act as reducing agents, capping agents, and stabilizing agents for the developed AgNPs. To date, several researchers have utilized potential bacteria for the formation of silver nanoparticles (AgNPs) by using either wet biomass or bacterial supernatant, for instance, wet biomass of *Micrococcus luteus* (*Vimalanathan et al., 2013*) and supernatant of the *Bacillus* ROM6 was used for the synthesis of AgNPs (*Esmail et al., 2022*). This particular bacterium was isolated from the Zarshouran gold mine in South Korea.

Other research studies include the synthesis of spherical-shaped AgNPs of size 5–50 nm using bacterial strains *E. coli*, *Exiguobacterium aurantiacumm*, and *Brevundimonas diminuta* (*Saeed, Iqbal & Ashraf, 2020*) and assessed the antimicrobial activity of the synthesized AgNPs against methicillin-resistant *Staphylococcus aureus* (MRSA) and several other multiple drug resistance (MDR) bacteria. Morphologically distinct types of AgNPs synthesized using different *Geobacillus* bacteria strains, namely 18, 25, 95, and 612, were also reported (*Cekuolyte et al., 2023*).

Similarly, the studies conducted by *Srinivasan et al. (2022)* reported the synthesis of AgNPs using a bioluminescent bacterium (*Vibrio campbellii*) and assessment of antibacterial and antioxidant properties of these synthesized AgNPs (*Srinivasan et al., 2022*). It would be beneficial to conduct a study to examine how various bacteria impact the morphology and purity of AgNPs. Furthermore, it is crucial to investigate the effect of morphologically diverse AgNPs on pathogenic bacteria regarding their antimicrobial properties (*Raza et al., 2021*). *Gola et al. (2021)* synthesized 6–25 nm, spherical, and

hexagonal-shaped AgNPs from *Aspergillus* sp. and used them to remove reactive yellow dye and antibacterial potential activity (*Gola et al., 2021*). *Rasheed et al. (2023)* synthesized AgNPs from *Conocarpus erectus* and *Pseudomonas* sp. and applied them to the elimination of reactive black 5 (RR5) and reactive red 120 (RR120) from the aqueous solutions.

From all the above investigations, various gaps were found, which are mentioned below: (i) the effect of different bacterial strains and their enzymes on the morphology of the synthesis of AgNPs; (ii) the utilization of AgNPs along with composite material to form nanocomposite for antimicrobial properties could not reveal the effectiveness of the antimicrobial properties of AgNPs in the composite. So, to understand these two issues, as mentioned earlier, a detailed investigation is needed to study the effect on the morphology of AgNPs and their purity by different bacteria and their enzymes, as well as the antimicrobial potential effect of morphologically diverse AgNPs on the pathogenic Gram-positive bacteria (GPB), and Gram-negative bacteria (GNB).

The current research aimed to synthesize silver nanoparticles from different bacteria (*Klebsiella pneumoniae, Micrococcus luteus*, and *Enterobacter aerogenes*) and compare their morphological, chemical, and elemental properties. One of the objectives was to confirm the formation of AgNPs, along with purity and morphology, by using Fourier transform-infrared (FTR-IR), UV-Vis spectrophotometer (UV-Vis), X-ray diffraction pattern (XRD), field emission scanning electron microscopy (FESEM), and energy dispersive X-ray spectroscopy (EDS). Another objective was to observe the morphological and elemental diversity among the bacterially synthesized AgNPs. Another objective was to evaluate the potential of AgNPs as an adsorbent for the removal of methyl orange (MO) dye from aqueous solutions. The final objective was to assess the potential of the synthesized AgNPs as an antibacterial agent against GPB *Bacillus subtilis* (MBC23), *B. cereus* (MBC24), and *B. megaterium* (MBP11) and GNB *Enterococcus faecalis* (MBP13).

# MATERIALS AND METHODS

## Materials

*Klebsiella pneumoniae* (PME2), *K. pneumoniae* (MBC34), *Micrococcus luteus* (MBC23), *Micrococcus luteus* (MBC57), *Enterobacter aerogenes* (MBX6), *Bacillus subtilis* (MBC3), *Bacillus cereus* (MBC4), *Bacillus megaterium* (MBP11), *Enterococcus faecalis* (MBP13), and *Serratia marcescens* (MBC27) were obtained as gift samples from the Gujarat Biotech Research Centre (Gandhinagar, Gujarat, India). The materials used in the study included silver nitrate (SRL, Gujarat, India), nutrient agar media, nutrient broth and antibiotic assay media (Himedia, Mumbai, India); ethanol (SLC Chemicals, Delhi, India), Whatman filter paper no. 42. (Axiva, Mumbai, India); and methyl orange (MO) (Loba, Chemie, Gujarat, India) and double distilled water (ddw). All the chemicals were analytical grade except silver nitrate and methyl orange dye (LR grade).

## Methods

### Screening and selection of bacteria for the synthesis of AgNPs

Around 10 bacterial species namely *Klebsiella pneumoniae* (PME2), *K. pneumoniae* (MBC34), *Micrococcus luteus* (MBC23), *Micrococcus luteus* (MBC57), *Enterobacter*

*aerogenes* (MBX6), *Bacillus subtilis* (MBC3), *Bacillus cereus* (MBC4), *Bacillus megaterium* (MBP11), and *Enterococcus faecalis* (MBP13), *Serratia marcescens* (MBC27), on the nutrient agar Petri plates were obtained from GBRC, which were later stored in a refrigerator in the laboratory. Further, about 200 mL of nutrient broth was prepared, to which about 1 mM of an aqueous solution of AgNO₃ was added. Additionally, about 10 mL of this mixture was transferred into 10 different Erlenmeyer flasks of 50 mL. A loopful culture of each bacterial strain was added to all these flasks and incubated in an incubator shaker at 37 °C for 2–3 days. After incubation, a color change was noticed, and later on, UV-Vis spectra were taken for all the samples. Out of all these bacterial strains, only three were found positive for the AgNPs synthesis, namely, *K. pneumoniae* (MBC34), *M. luteus* (MBC23), and *E. aerogenes* (MBX6) as the color changed to red with a higher color intensity within 24–48 h. Further, an absorbance peak was obtained by UV-Vis analysis near 500–540 nm in all three samples. So further, only these three bacterial strains (*K. pneumoniae* (MBC34), *M. luteus* (MBC23), and *E. aerogenes* (MBX6)) were used for a large amount of AgNPs formation, which was previously identified by GBRC by using 16s rRNA genome sequencing.

### Synthesis of silver nanoparticles from bacteria

For the fabrication of AgNPs, an aqueous solution of silver ions was exposed to the bacterial supernatants obtained from all three bacterial strains under optimized conditions. For the mass production of AgNPs, all three bacterial strains were grown in a nutrient broth medium followed by inoculation with (*K. pneumoniae* (MBC34), *M. luteus* (MBC23), and *E. aerogenes* (MBX6)) individually in an Erlenmeyer flasks and incubation in an incubator shaker at 37 °C for 24 h at 120 rpm. Further, after 24 h, all three flasks were taken out, and the medium was transferred to centrifuge tubes followed by centrifugation at 5,000 rpm for 10 min. The bacterial supernatant was retained, while the bacterial pellet was discarded. Further, about 100 mL of all three bacterial supernatants (*K. pneumoniae* (MBC34), *M. luteus* (MBC23), and *E. aerogenes* (MBX6)) were taken separately in three amber bottles, and to each bottle having about 100 mL of silver nitrate solution was added. All three flasks and one control (silver nitrate solution + 100 mL ddw) were kept under dark and static conditions for 2–3 days, and color change was continuously monitored. Initially, the color of the aqueous solution of the silver nitrate was pale/transparent, but after the addition of bacterial supernatant, the color tuned to a milky white appearance. Finally, after 2–3 days, the color of the three bottles changed from milky white to reddish brown, indicating the formation of AgNPs. The mixture from each bottle was transferred to the centrifugation tubes separately and centrifuged at 5,000 rpm for 10 min. The supernatant was discarded, while the solid particle having AgNPs was retained. Further, all three AgNPs were washed 2–3 times with ddw and once with ethanol. AgNPs powders were then transferred to three different Petri plates and kept for drying in an oven at 50–60 °C until complete dryness.

### Preparation of aqueous solution methyl orange (MO) dye

A 50-ppm aqueous solution of MO dye was prepared by adding 50 mg of MO dye powder granules into the 1,000 mL ddw. The aqueous solution was kept on a magnetic stirrer with vigorous stirring at 250 rpm to dissolve the dye granules completely. Further, Whatman filter paper was used for the filtration of the aqueous solution to eliminate the impurities. Finally, the dye sample was placed in an amber-colored glass reagent bottle for future use.

### Batch study of adsorption of MO dye

About 100 mL of an aqueous solution of MO dye was taken from the stock solution into three different glass beakers of appropriate volume. All three glass beakers were placed on a magnetic stirrer, and 1 mg of AgNPs of each type was added to separate glass beakers. The interaction between the AgNPs and MO dye was carried out by agitation at 400 rpm for all three experimental flasks and one control (50 ppm MO dye, no AgNPs). Further, an aliquot (2–3 mL) was collected from all four glass beakers at 0, 30, 60, 90, and 120 min. All the collected samples were then analyzed by the UV-Vis spectrophotometer to identify the concentration of the dye samples. The UV-Vis absorbance maxima of MO dye are 520 $\pm$ 15 nm. Further, the MO dye removal percentage was measured by using the following formula as provided by *Swathilakshmi et al. (2022)* in Eq. (1):

$$\% \ Dye \ removal = \frac{Co - Ct}{Co} \times 100 \tag{1}$$

where,

$C_o$ = initial dye concentration,

$C_t$ = dye concentration at a specific time.

### Antimicrobial activity of the synthesized AgNPs

The antimicrobial properties of all the bacterially developed AgNPs (AgNPs-K, AgNPs-M, and AgNPs-E) were assessed against GPB: *B. subtilis, B. cereus, B. megaterium*, and GNB: *E. faecalis* by the disc diffusion method (*Yassin et al., 2022*). Firstly, 16 discs of a specific size (8 mm diameter) were cut out of Whatman filter paper and stored in a glass vial. Further 5 mg solution of all three AgNPs was prepared by dispersing in ddw. Further, four discs were added into each type of AgNPs vial *i.e.*, AgNPs-K, AgNPs-M, and AgNPs-E. Further, all the reagent vials were sonicated for 15–20 min using an ultrasonicator (Lequitron). Additionally, the AgNPs loaded discs were taken out of the vials with the help of forceps kept on three different Petri plates and dried in a hot air oven at 40–50 °C. Further, four different antibiotic assay media were prepared. Further, each plate was inoculated with different *Bacillus* spp., *B. subtilis* (MBC23), *B. cereus* (MBC24), *B. megaterium* (MBP11), and *E. faecalis* (MBP13) by using a sterilized cotton swab under aseptic conditions. All the three types of dried AgNPs loaded filter paper discs were gently placed on each bacteria-swabbed Petri plate. Finally, the Petri plates were incubated in a bacterial incubator at 37 °C, and after 24 h, plates were observed for the evaluation of the antimicrobial activity of the synthesized AgNPs. The ZOI was measured using a measurement scale against the light, and the size was recorded in mm (*Ballén et al., 2021*).

## CHARACTERIZATION OF SILVER NANOPARTICLES

### UV-Visible spectroscopy

The UV-Vis measurement of all three types of AgNPs was done by adding 1 mg of AgNPs in 5 mL of ddw in three different test tubes, followed by sonication for 10 in an ultrasonicator. An aliquot (2 mL) was transferred to the quartz cuvette from the well-dispersed AgNPs samples, and the UV-Vis measurement was done in the range of 200–800 nm at a resolution of 1 nm by using a UV-Vis spectrophotometer (UV 1800; Shimadzu spectrophotometer, Shimadzu, Japan). The UV-vis measurement of the control sample was also done in the above range by using a double-beam LMSP-UV100S (Labman, Chennai, India), instrument.

### FTIR

The FTIR measurement was done to identify the various functional groups present in the bacterially synthesized AgNPs. The FTIR measurement was done using the solid KBr pellet method, where the pellets were prepared by mixing 2 mg AgNPs and 198 mg KBr for all three types of AgNPs. The measurement was done in the mid-IR region 599–4,000 cm$^{-1}$ at a resolution of 2 cm$^{-1}$ by using a spectrum S6500 instrument (Perkin-Elmer, Waltham, MA, USA).

### XRD

The XRD patterns for all three types of AgNPs samples were recorded using a Miniflex 800 (Rigaku, Houten, Netherlands) instrument equipped with an X'celerometer to reveal the crystallinity. XRD patterns were recorded in the 2-theta range of 20–70 by using a filter K-beta (x1) with a step size of 0.02 and a time of 5 s per step, scan speed/duration time: 10.0 degree/min., step width: 0.0200 degrees at 30 kV voltage, and a current of 2 mA.

### FESEM-EDS

The morphological analysis of all three types of AgNPs was investigated using a FEI Nova NanoSEM 450 (USA). The dry AgNPs were loaded on the carbon tape with the help of a fine brush, which in turn was kept on the Al stub holder. All the samples were exposed to gold sputtering. The elemental analysis of AgNPs was carried out using an Oxford-made energy-dispersive X-ray spectroscopy (EDS) analyzer fixed to the FESEM at variable magnifications at 20 kV.

## RESULTS AND DISCUSSION

### Mechanism of formation of AgNPs by bacteria

The bacterial strains, *i.e.*, *K. pneumoniae* (MBC34), *M. luteus* (MBC23), and *E. aerogenes*, have numerous microbial proteins and enzymes that help in the reduction of Ag$^+$ ions into Ag$^0$ (*Ballén et al., 2021*). The actual mechanism of the biosynthesis of AgNPs by bacteria is well described in the literature. It is a simple mechanism where the oxidized silver ions get two electrons from microbial proteins and enzymes and are reduced to the stabilized Ag$^0$. When the AgNO$_3$ aqueous solutions are mixed with the bacterial culture/supernatant, the bacterial enzymes present in the supernatant reduce the Ag$^+$ ions into Ag$^0$. So, during this

step, the previously milky color of the aqueous silver solutions gets converted to red, indicating the development of AgNPs in the medium. Moreover, these biomolecules may also act as stabilizing and capping agents for the synthesized AgNPs (*Giri et al., 2022*; *Terzioğlu et al., 2022*). Figure 1 shows the mechanism involved in the formation of AgNPs from the bacteria. Here, the color of the medium changed from milky white to dark brown within 2 to 3 days. Earlier, a similar color change (yellow to brown) was observed during the synthesis of AgNPs by *K. pneumoniae* isolated from humans and sheep (*Sayyid & Zghair, 2021*). *Saleh & Khoman Alwan (2020)* used *K. pneumoniae* culture supernatant for the biosynthesis of AgNPs. Earlier, *Javaid et al. (2018)* also suggested a similar pathway for the formation of AgNPs from the bacteria *via* a NADH-dependent nitrate reductase enzyme. Table 1 shows the major microbial proteins and enzymes present in *K. pneumoniae, M. luteus*, and *E. aerogenes*.

## UV-Vis analysis for preliminary confirmation of the formation of AgNPs

Figure 2 shows a typical absorbance peak in the range of 425–505 nm for all three AgNPs synthesized by bacteria by UV-Vis instrument, indicating the formation of AgNPs (*Saleh & Khoman Alwan, 2020*). Earlier, *Saleh & Khoman Alwan (2020)* obtained a peak at 432 nm for the AgNPs synthesized by *K. pneumoniae* (*Saleh & Khoman Alwan, 2020*), 420 and 440 nm by *M. luteus* (*Vimalanathan et al., 2013*), 450 nm by cyanobacterium *Oscillatoria limnetica* (*Hamouda et al., 2019*), 450 nm by endophytic bacteria *Enterobacter roggenkampii* BLS02 (*Kumar & Dubey, 2022*), and 405–407 nm for *Klebsiella pneumoniae* by *Kalpana & Lee (2013)*. The AgNPs formed only when the $AgNO_3$ to bacterial supernatant ratio was about 4:6, during which there was higher absorption intensity. Moreover, the amount of bacterial culture filtrate and reducing agents in the supernatant governs the yield of the AgNPs (*Kalpana & Lee, 2013*).

## Identification of functional groups of silver NPs by FTIR

A typical FTIR spectra of all the AgNPs (AgNPs-K, AgNPs-M, and AgNPs-E) synthesized by bacteria is shown in Fig. 3, which was used for the identification of various functional groups present in AgNPs. All the samples have a common band at 599 $cm^{-1}$ and 963, 1,299, 1,349, 1,693, 2,299, 2,891, and 3,780 $cm^{-1}$. The band at 599 $cm^{-1}$ is attributed to the metallic Ag. The band at 963 $cm^{-1}$ is attributed to the amide V band arising due to out-of-plane NH bending of peptide linkages (*Kalpana & Lee, 2013*). A small intensity band at 1,051 $cm^{-1}$ is attributed to the primary amine C–N stretch. Another small intensity band at 1,349 $cm^{-1}$ in all the samples is attributed to the C-C bond. A small intensity band in all the samples at 1,699 $cm^{-1}$ is attributed to the OH group in the samples. Moreover, this band is also attributed to the C=O stretching of amide I bands of peptide linkage. The band at 1,229 $cm^{-1}$ is attributed to the CN stretching of peptide linkage. The band at 1,349 $cm^{-1}$ is attributed to the (C–C) stretching vibration of aliphatic amines, as previously documented by *Ibrahim et al. (2019)*. In this study, AgNPs were developed from endophytic bacteria, which showed a band at 1,359 $cm^{-1}$ (*Ibrahim et al., 2019*). All the samples exhibit an atmospheric carbon band at 2,891 $cm^{-1}$ is attributed to the methylene C–H asymmetric or

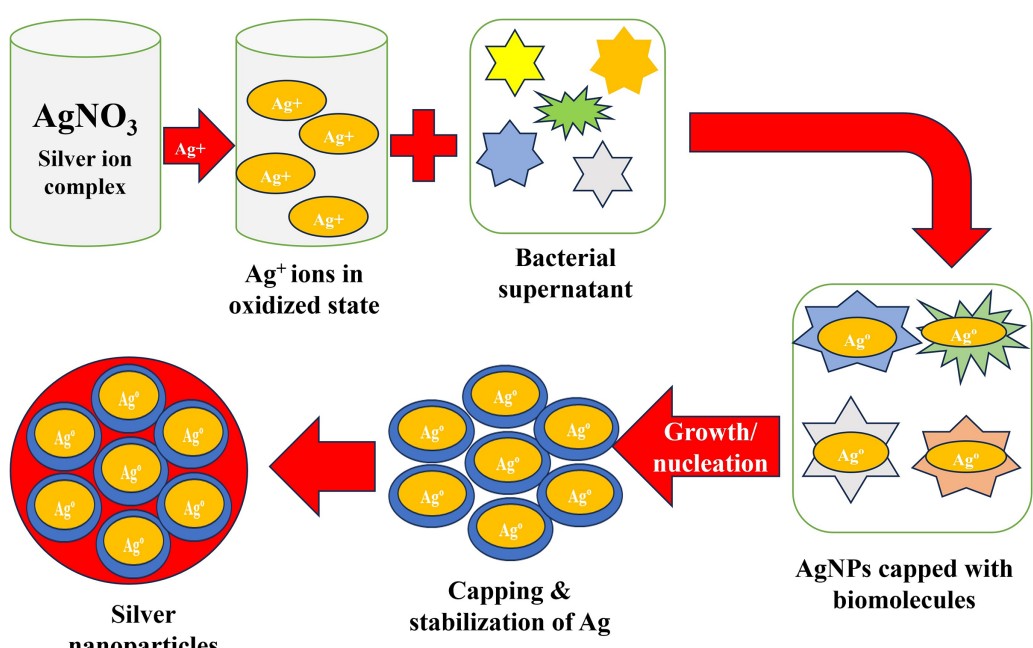

**Figure 1 Schematic diagram for the development of AgNPs from silver ions by bacteria.** The oxidized silver ions interact with the bacterial enzyme, leading to the reduction of the Ag⁺ ions. The silver ions get capped with biomolecules from the bacteria. Further, these silver ions grow and nucleate, which get capped and stabilized by bacterial biomolecules, leading to the formation of stabilized AgNPs.

**Table 1 The major microbial proteins and enzymes present in _K. pneumoniae_, _M. luteus_, and _E. aerogenes_.**

**Various microbial proteins and enzymes**

| _K. pneumoniae_ | _M. luteus_ | _E. aerogenes_ | References |
|---|---|---|---|
| _α-Lactamase_ | Esterase | _α-Lactamase_ | _Wintachai et al. (2020)_, _Karami-Zarandi et al. (2023)_ |
| _Lipases_ | Proteases | _Lipases_ | _Merciecca et al. (2022)_ |
| _Proteases_ | Phytases | _Proteases_ | _Wintachai et al. (2020)_ |
| _Amylases_ | Dehydrogenases | _Amylases_ | _Pan et al. (2020)_ |
| _Catalase_ | Lipases | _Catalase_ | _Taylor & Achanzar (1972)_ |
| _Gelatinase_ | – | _Gelatinase_ | _Austin et al. (2020)_ |
| _Urease_ | – | _Urease_ | _Carter et al. (2011)_ |

symmetric stretch. All the samples showing a small band from 3,400 to 3,800 cm⁻¹ centered at 3,780 cm⁻¹ are attributed to the -OH molecule.

Earlier, _Saleh & Khoman Alwan (2020)_ obtained four distinct peaks for the AgNPs synthesized by _K. pneumoniae_ at 3,332.78 cm⁻¹, 2,115.35, 1,635.60, and 1,096.92 cm⁻¹ and suggested that the band at 3,332.78 cm⁻¹ is due to the stretching vibration of the OH bond of alcohol and phenols. The band at 2,115.35 cm⁻¹ is attributed to the C-H stretching of the methylene groups of protein and the N-H stretching of amine salt. The band at 1,635.60 cm⁻¹ was attributed to the carbonyl groups (C=O) of the amino acid residues. In contrast,

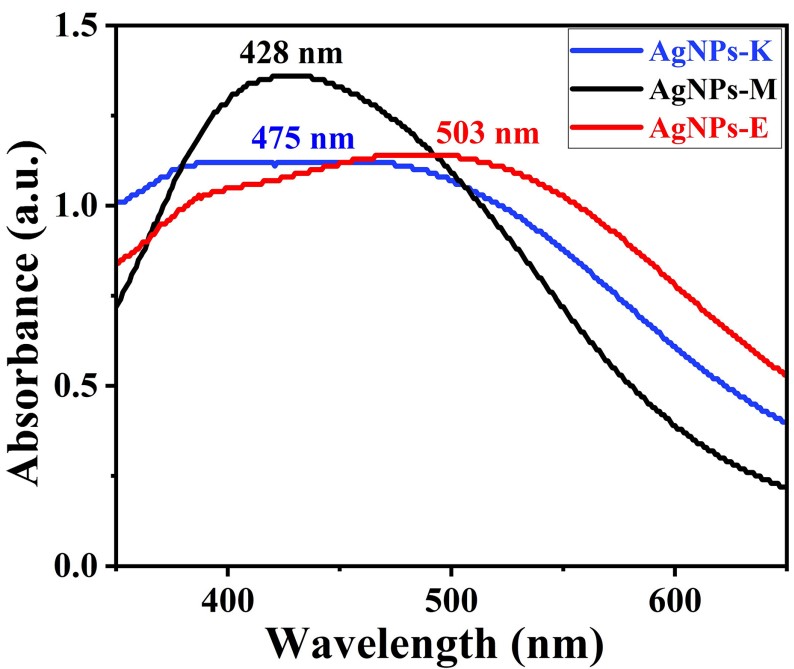

**Figure 2** UV-Vis measurement of different AgNPs synthesized by bacteria.

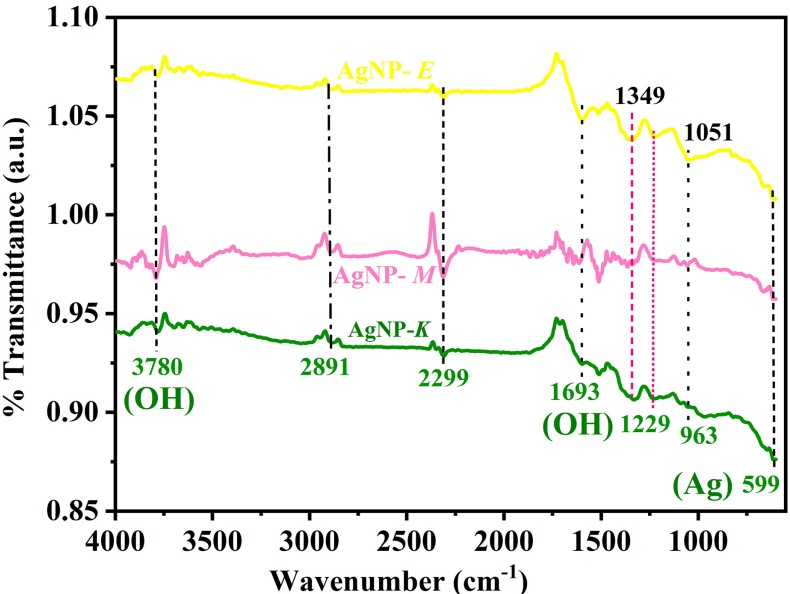

**Figure 3** FTIR spectra of AgNPs synthesized by bacteria. The bands at 599 and 963 cm$^{-1}$ represent Ag, 1,229 cm$^{-1}$ represent C≡N stretching of peptide linkage, 1,693 and 3,780 cm$^{-1}$ represent OH, and 2,891 cm$^{-1}$ represent atmospheric carbon band.

the band at 1,096.92 cm$^{-1}$ was attributed to the (C-O) stretching of alcohols and esters, carboxylic acids, and the C–N stretching of aliphatic amines (*Saleh & Khoman Alwan, 2020*). Based on the above information, it was further concluded that the presence of

protein in the supernatant acts as a stabilizing and capping agent for stabilization, which binds to the synthesized AgNPs through free cysteine or amine groups in proteins (*Saleh & Khoman Alwan, 2020*). The results were in close agreement with the investigation carried out by *Kalpana & Lee (2013)* where major intensity bands were obtained at 2,964.55, 1,262.22, 1,095.89, 1,021.96, 800.73 cm$^{-1}$, and small intensity bands at 2,960.64, 1,650.01, 865.33, 701 and 477.07 cm$^{-1}$ for the AgNPs synthesized by *K. pneumoniae.*

In the current investigation, authors have also obtained bands for the *K. pneumoniae* synthesized AgNPs at 599, 963, 1,229, 1,693, 2,891, and 3,780 cm$^{-1}$, which correspond to the bands obtained by *Kalpana & Lee (2013)*. Similarly, in another study also, four distinct FTIR bands were observed at 1,643, 1,586, 1,397, and 1,042 cm$^{-1}$, and it was concluded that the AgNPs synthesized by bacteria exhibited improved stability because the coating of AgNPs by bacterial and media components (*Peiris et al., 2018*). AgNPs-M and AgNPs-E displayed similar bands as those of AgNPs-K with slight variation in their intensity.

## Phase identification of silver nanoparticles by XRD

The XRD investigation was carried out to identify the crystalline phase of the AgNPs. A typical XRD pattern of all the bacterially synthesized AgNPs is shown in Fig. 4, where all the AgNPs exhibit three characteristic peaks of silver NPs at 27.6°, 32.1°, and 46.2°, and three small intensity peaks at 54.7°, 57.4°, and 76.7°. The major intensity peaks in all three types of bacterially synthesized AgNPs are at 32.1°, followed by 46.2° and 27.6°. The XRD planes in all three types of AgNPs were 101, 111, 200, 220, and 311, as matched with the Joint Committee on Power Diffraction Standards (JCPDS) 03-0921. The diffraction peaks obtained in the current investigation at 27.6°, 32.1°, and 46.2° and small diffraction peaks at 54.7°, 57.4°, and 76.7° for the AgNPs-K. AgNPs-M also has the same peaks as AgNPs-K, *i.e.*, at 27.6°, 32.1, 46.2, 54.7, and 57.4°. All the XRD peaks for AgNPs-M were of almost the same intensity except for 27.6°, which was comparatively stronger than AgNPs-K. The XRD peaks for AgNPs-E were also at the same places as those of AgNPs-M and AgNPs-K, but the peak at 27.6° was of stronger intensity than AgNPs-K and weaker than AgNPs-M. Moreover, the peak at 46.2° in both AgNPs-M and AgNPs-E was stronger than the AgNPs-K. Further, the peaks at 54.7° (311) and 57.4° (222) were like a small, broad hump in both AgNPs-K and AgNPs-M, but these two peaks were sharper in AgNPs-E.

The results were in close agreement with the previous results obtained by *Kalpana & Lee (2013)* and *Ibrahim et al. (2019)*, where *Kalpana & Lee (2013)* obtained diffraction peaks at 37.76°, 45.87°, 64.08°, and 77.11°, which were indicated by the (111), (200), (220), and (311) reflections of metallic Ag. The data obtained here was matched with the database of JCPDS file no. 03-0921 (*Kalpana & Lee, 2013*). *Ibrahim et al. (2019)* obtained diffraction peaks for the AgNPs synthesized by an endophytic bacteria, *Bacillus siamensis* strain C1, at 27.81, 32.34, 46.29, 57.47, and 77.69°, corresponding to (101), (111), (200), (220), and (311) crystal planes, respectively, for the AgNPs. Earlier, the study by *Vimalanathan et al. (2013)* also obtained similar results, where the major peaks were at 28, 32, and 47°, two small intensity peaks at 55° and 57°, and a small diffraction peak at 76°.

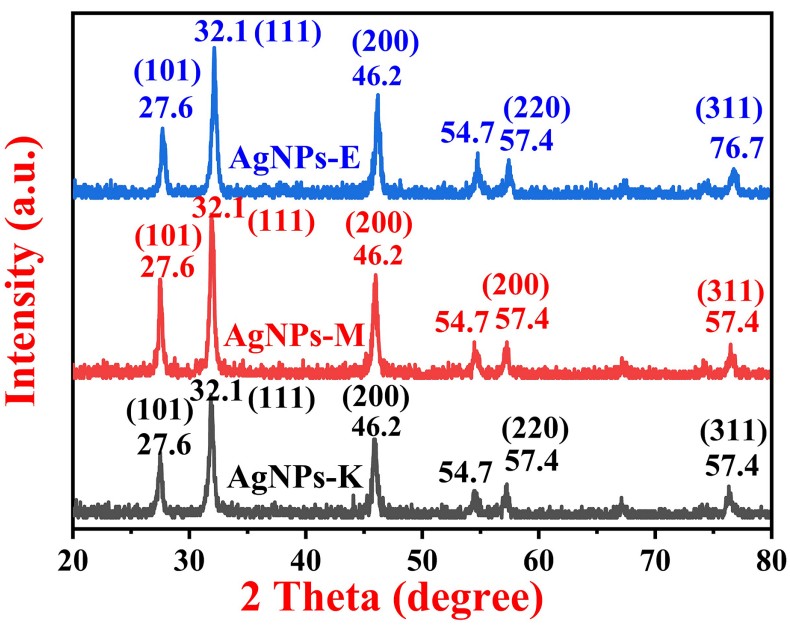

**Figure 4 XRD pattern of silver nanoparticles synthesized by different bacteria.**

The crystalline size of all three AgNPs synthesized by bacteria was determined by using the Scherrer formula as given in Eq. (2),

$$D = \frac{k\lambda}{\beta Cos\theta}$$  (2)

where,

D = crystalline size,

k = constant (0.9), and

β = the FWHM values of the diffracted peaks.

The highest intensity peak was used to find all the parameters in the Scherrer equation. The Gaussian peak fits were used to find the FWHM values and exact theta values. The calculated crystalline size was found to be around 16.88, 18.00, and 16.44 nm for AgNPs-K, AgNPs-M, and AgNPs-E. Therefore, it is well examined that the synthesized AgNPs showed a crystalline nature and 16.88, 18.00, and 16.44 nm crystallite sizes.

## Morphological analysis of silver nanoparticles by FESEM and elemental analysis by EDS

Figures 5A–5F shows FESEM micrographs of AgNPs synthesized by *K. pneumoniae* (AgNPs-K). Figures 5A and 5B show a porous flakes-like structure that is embedded with bright-colored AgNPs. Figures 5C and 5D clearly shows rhombohedral-shaped AgNPs, whose size varies from 22–66 nm. These two images clearly show that the AgNPs are embedded in porous, sheet-like structures. Figures 5E and 5F shows aggregated spherical-shaped structures. Previously, several investigators have demonstrated similar morphology for the bacterially synthesized AgNPs.

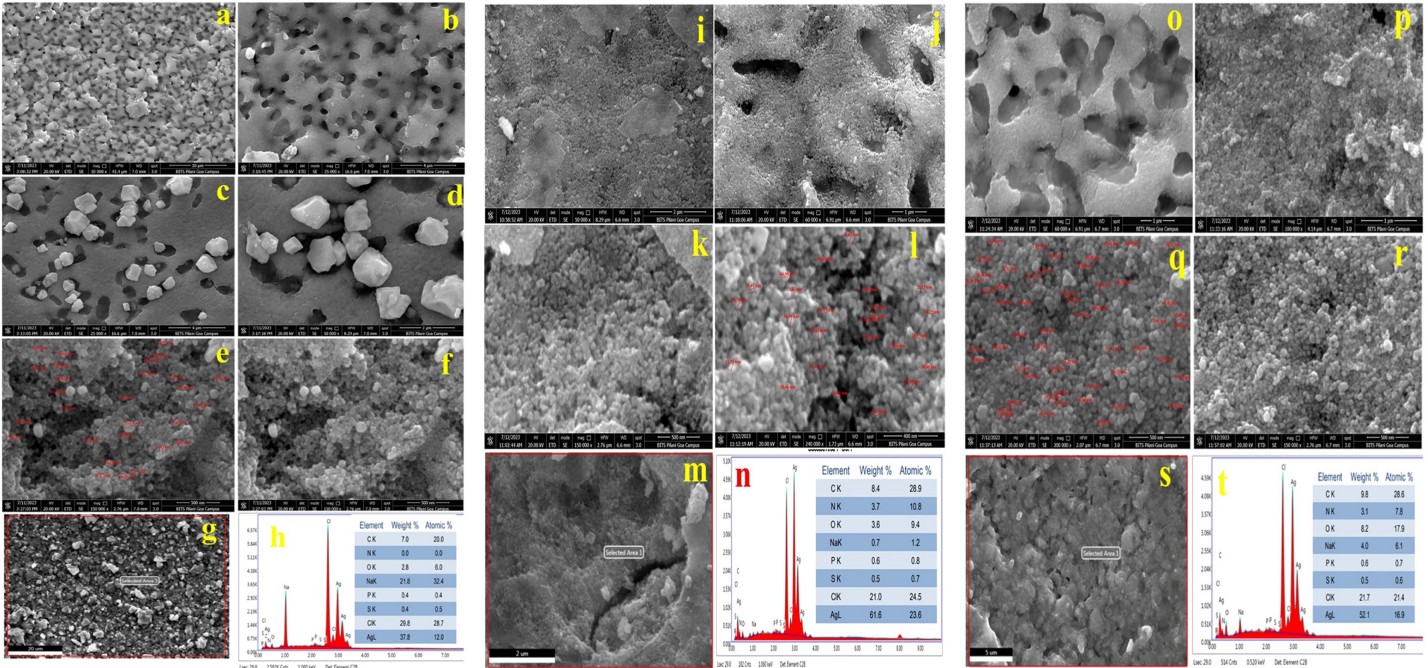

**Figure 5 FFESEM images (A–F), EDS spot (G), and EDS spectra and elemental table (H) for AgNPs-K.** FESEM images (I–L), EDS spot (M) and EDS spectra, and elemental table (N) for AgNPs-M. FESEM images (O–R), EDS spot (S) and EDS spectra, and elemental table (T) for AgNPs-E.

Figure 5G shows the EDS spot of the AgNPs-K, while Fig. 5H exhibits the EDS spectra and elemental table of the AgNPs-K. The EDS spectra in Fig. 5H show peaks for Ag, Cl, Na, P, S, C, O, and N. Among these, the elements contributing the most to the sample were Ag (37.8% At wt.), Cl (29.8%), and P and S 0.4% each. The P and S were not present in trace amounts in the AgNPs-K. The major impurities in the synthesized AgNPs-K are NaCl, which alone comprises 50% due to the improper washing of the samples. Moreover, these two may also come from the nutrient broth used for growing the bacteria. At the same time, the presence of C, S, and P indicates the association of biomolecules with the synthesized AgNPs-K. Earlier, *Sayyid & Zghair (2021)* reported cube-shaped to irregular heterogeneous forms of AgNPs synthesized by *K. pneumoniae*, whose average size was 40.47 nm (*Sayyid & Zghair, 2021*). Moreover, investigators further observed that the morphology by TEM was a pseudo-spherical shape of size 40–80 nm.

*Saleh & Khoman Alwan (2020)* obtained spherical-shaped particles of size 26.84 to 44.42 nm, which were highly aggregated. Further, the investigators concluded that the conglomeration of the AgNPs occurs during the drying process.

*Rasheed et al. (2023)* obtained nano-rod-like AgNPs of size 100–200 nm from the *Conocarpus* erectus plant, while oval-shaped AgNPs of size 110–150 nm were synthesized from *Pseudomonas* spp., and by applying the chemical reduction method, flower-like AgNPs of size 100–200 nm were obtained. So, the smallest size was reported from *Pseudomonas* spp., which was oval-shaped.

Figures 5I–5L shows FESEM micrographs of AgNPs synthesized by the *M. luteus* (AgNPs-M). Figures 5I and 5J shows a high aggregation of the synthesized AgNPs-M. Figures 5K and 5L shows spherical-shaped AgNPs-M, whose size varies from 21–45 nm. Figure 5M shows the EDS spot of the AgNPs-M, while Fig. 5N exhibits the EDS spectra and elemental table of the AgNPs-M. The EDS spectra of AgNPs-M in Fig. 5N show peaks for Ag, Cl, Na, P, S, C, O, and N. Out of all these, the significant elements were mainly Ag (61% At wt.), Cl (21%), C (8.4%), N (3.7%), and O (2.6%). Other detected elements, such as Na, P, and S, were present in trace amounts, and the major impurity in the final sample was Cl due to improper sample washing. Moreover, it also came from the bacterial media, *i.e.*, nutrient broth. At the same time, the presence of C, N, S, and P indicates the presence of biomolecules with AgNPs-M.

Figures 5O–5R shows FESEM micrographs of AgNPs synthesized by *E. aerogenes* (AgNPs-E). Figures 5O and 5P shows a porous flakes-like structure embedded with the bright color of AgNPs. Figures 5Q and 5R clearly shows rhombohedral-shaped AgNPs-E, whose size varies from 24–60. The images clearly show that the AgNPs-E are embedded in the porous flakes-like structures. The particles are showing high aggregation, as evident from the SEM micrographs. Figure 5S shows the EDS spot of the AgNPs-E, while Fig. 5T exhibits the EDS spectra and elemental table of the AgNPs-E. Figure 5T shows the spectra of Ag, Cl, Na, P, S, C, O, and N. Out of all these, the significant elements were mainly Ag (52.1% At wt.), Cl (21.7%), C (9.8%), O (8.2%), Na (4.0%), and N (3.1%). In addition to this, P and S were present in trace amounts, whose total composition was near 1.1%. The major impurities in the synthesized AgNPs-E are NaCl, which alone comprises 25.7%, which indicates the improper washing of the sample. Moreover, these two may also come from the nutrient broth used for growing the bacteria. The presence of C, S, N, and P indicates the association of enzymes and proteins from the *E. aerogenes* with the synthesized AgNPs-E. Table 2 shows the major elements present in all three types of AgNPs synthesized by bacteria.

From the EDS data of all three types of AgNPs, it was found that Ag was present in the highest percentage in AgNPs-M and at least 37.8% in AgNPs-K. Among all the three types of AgNPs, Cl was present most in AgNPs-K and least in AgNPs-M (21.0%). The oxygen was present in the highest amount in AgNPs-E and the least in AgNPs-K, *i.e.*, 8.2% and 2.8%, respectively. The carbon was highest in AgNPs-E (9.8%) and least 7.0% in AgNPs-K. Out of all the three types of AgNPs, Na was highest in AgNPs-K (21.8%) and least in AgNPs-M (0.4%). Among all the three samples of AgNPs, N was present highest in AgNPs-M (3.7%) and 3.1% in AgNPs-E, and it was not detected in AgNPs-K. The P and S were present almost identically in all the samples but least in AgNPs-K, *i.e.*, 0.4%.

In the current investigation, the authors found a broad peak of silver ions at 3 keV in all three types of AgNPs, which confirmed the reduction of $Ag^+$ to $Ag^0$. Moreover, here, the authors have mainly considered peaks in EDS for Ag, Cl, and carbon. The peaks for Ag, Cl, and S were consistent with the results obtained for the AgNPs synthesized by endophytic bacteria by Ibrahim and his team. *Ibrahim et al. (2019)* and his research group also concluded that the broad peak of silver ions was formed at 3 keV, which indicated the

**Table 2 Comparison between all the elements present in all three types of AgNPs.**

| Elements (At. wt.%) | AgNPs-*M. luteus* | AgNPs-*K. pneumoniae* | AgNPs-*E. aerogenes* |
|---|---|---|---|
| Ag | 61.6 | 37.8 | 52.1 |
| Cl | 21.0 | 29.8 | 21.7 |
| O | 3.6 | 2.8 | 8.2 |
| C | 8.4 | 7.0 | 9.8 |
| Na | 0.7 | 21.8 | 4.0 |
| N | 3.7 | 0.0 | 3.1 |
| P | 0.6 | 0.4 | 0.6 |
| S | 0.5 | 0.4 | 0.5 |

reduction of $Ag^+$ to $Ag^0$. Table 3 shows the comparative analysis of all the previously reported bacterially synthesized AgNPs with the current investigation.

From all the previous investigations, it was revealed that the largest size of AgNPs was 40.47 ± 89 nm synthesized by using *K. pneumoniae*, whose shape was cube to spherical and, AgNPs synthesized by *B. cereus*, was smallest in size, *i.e.*, 2–16 nm with spherical in shape. In the current investigation, the size of the synthesized AgNPs varied from 21 to 66 nm. Previously, three separate research studies reported on synthesized AgNPs from different species of *K. pneumoniae*, which were mainly spherical and cube-shaped. In most cases, the synthesized AgNPs had a spherical shape, with only a few exceptions, and in two instances, cube-shaped AgNPs were obtained. Regarding elemental composition and purity, AgNPs synthesized by *B. siamensis* strain C1 were purest, where Ag was 91.8%. At the same time, the remaining was impurity mainly by Cl and S, whereas in our case, the Ag (%) varied from 37.8% to 61.6%, *i.e.*, less than reported previously (*Ibrahim et al., 2021*). The lower purity could be due to improper washing of the AgNPs during centrifugation. From the UV-Vis study, it was found that the absorbance peak of AgNPs synthesized from different bacteria could be varied from 400 to 510 nm based on the size and shape of the synthesized AgNPs. From XRD and FTIR, it was found that the majority of the peaks and bands remain the same in all the syntheses with slight variations, respectively.

## Batch adsorption study of MO dye by AgNPs

MO dye shows the highest absorbance at 464 nm when examined using a UV-Vis spectrophotometer. The AgNPs-M, AgNPs-K, and AgNPs-E treated MO were measured for their color intensity in an aqueous dye solution for up to 120 min at a regular interval of 30 min. With time, the concentration of MO dye in the sample decreased gradually, evident from the UV-visible spectra (Figs. 6A–6C). So, the maximum removal of MO dye was found after 120 min using AgNPs-M. The analysis at an interval of every 30 min shows a gradual decrease in the concentration and absorbance of the dye; hence, the reduction in the graph can be observed easily as it progresses from 0 to 120 min. Figures 6A–6C show the MO dye absorbance by UV-Vis spectroscopy at different time intervals. Figure 6A

**Table 3 The comparative analysis of all the previous studies and current investigations of the bacterially synthesized AgNPs.**

| Microorganism | Particle size (nm) | Elements observed | Morphology | XRD peaks and size | Absorbance maxima wavelength (nm) | FTIR bands | References |
|---|---|---|---|---|---|---|---|
| K. pneumoniae | 40.47 ± 89 | | Cube-shape, irregular heterogeneous forms | – | – | – | Sayyid & Zghair (2021) |
| Pseudomonas aeruginosa ATCC 27853 | 11.71 ± 2.73 | – | Spherical | – | 427 | 1,643, 1,586, 1,397 and 1,042 cm$^{-1}$ | Peiris et al. (2018) |
| S. aureus ATCC 25923 | 11.14 ± 6.59 | – | | – | 430 | 2,164.86, 2,049.17 and 1,979.69 cm$^{-1}$ | |
| E. coli ATCC 25922 | 13–16 | – | | | 420–435 | 2,916.88 cm$^{-1}$ | |
| Acinetobacter baumannii | 8–12 | – | | – | 420–435 | 1,643.75, 2,161.48, and 2,924.28 cm$^{-1}$ | |
| K. pneumoniae | 5 & 50 (TEM) | – | Spherical, & triangular (small amount) | 37.76°, 45.87°, 64.08° and 77.11° for e (1 1 1), (2 0 0), (2 2 0) and (3 1 1) | 405–407 | 2,964.55, 1,262.22, 1,095.89, 1,021.96, 800.73, 2,960.64, 1,650.01, 865.33, 701, 477.07 cm$^{-1}$ | Kalpana & Lee (2013) |
| K. pneumoniae | 26.84 to 44.42 | – | Cube-shape to heterogenous, agglomerated | – | 432 | 2,115.35, 1,635.60, 3,332.78 & 1,096.92 cm$^{-1}$ | Saleh & Khoman Alwan (2020) |
| B. siamensis strain C1 | 25 to 50 Average: 34 ± 3 | Ag (91.8%), Cl (7.5%), S (0.7%) | Spherical | 27.81 (101), 32.34 (111), 46.29 (200), 57.47 (220), & 77.69 (311) | 409 | 3,385, 2,925, 1,732, 1,645, 1,556, 1,359, 1,079 & 537 cm$^{-1}$ | Ibrahim et al. (2019) |
| B. cereus | 2–16 | | Spherical | | 420 | | Gurunathan (2019) |
| K. pneumoniae | 22–66 | Ag, Cl, C, Na, O, S, N & P | Spherical to irregular | 27.6, 32.1, 46.2, 54.7, 57.4% 76.7 Size: 16.88 nm | 475 | 599, 963, 1,299, 1,349, 1,693, 2,299, 2,891, & 3,780 cm$^{-1}$ | Current investigation |
| M. luteus | 21–45 | Ag, Cl, C, Na, O, S, N & P | Spherical to irregular | 27.6, 32.1, 46.2, 54.7, 57.4% 76.7 Size: 18 nm | 503 | 599, 963, 1,299, 1,349, 1,693, 2,299, 2,891, & 3,780 cm$^{-1}$ | Current investigation |
| E. aerogenes | 24–60 | Ag, Cl, C, Na, O, S, N & P | Spherical to irregular | 27.6, 32.1, 46.2, 54.7, 57.4% 76.7 Size: 16.44 nm | 428 | 599, 963, 1,299, 1,349, 1,693, 2,299, 2,891, & 3,780 cm$^{-1}$ | Current investigation |

(AgNPs-K), Fig. 6B (AgNPs-M), Fig. 6C (AgNPs-E), and Fig. 6D (control). The control flask had 50 ppm dye and no AgNPs. The UV-Vis spectra show no change in absorbance peaks from 0 to 120 min.

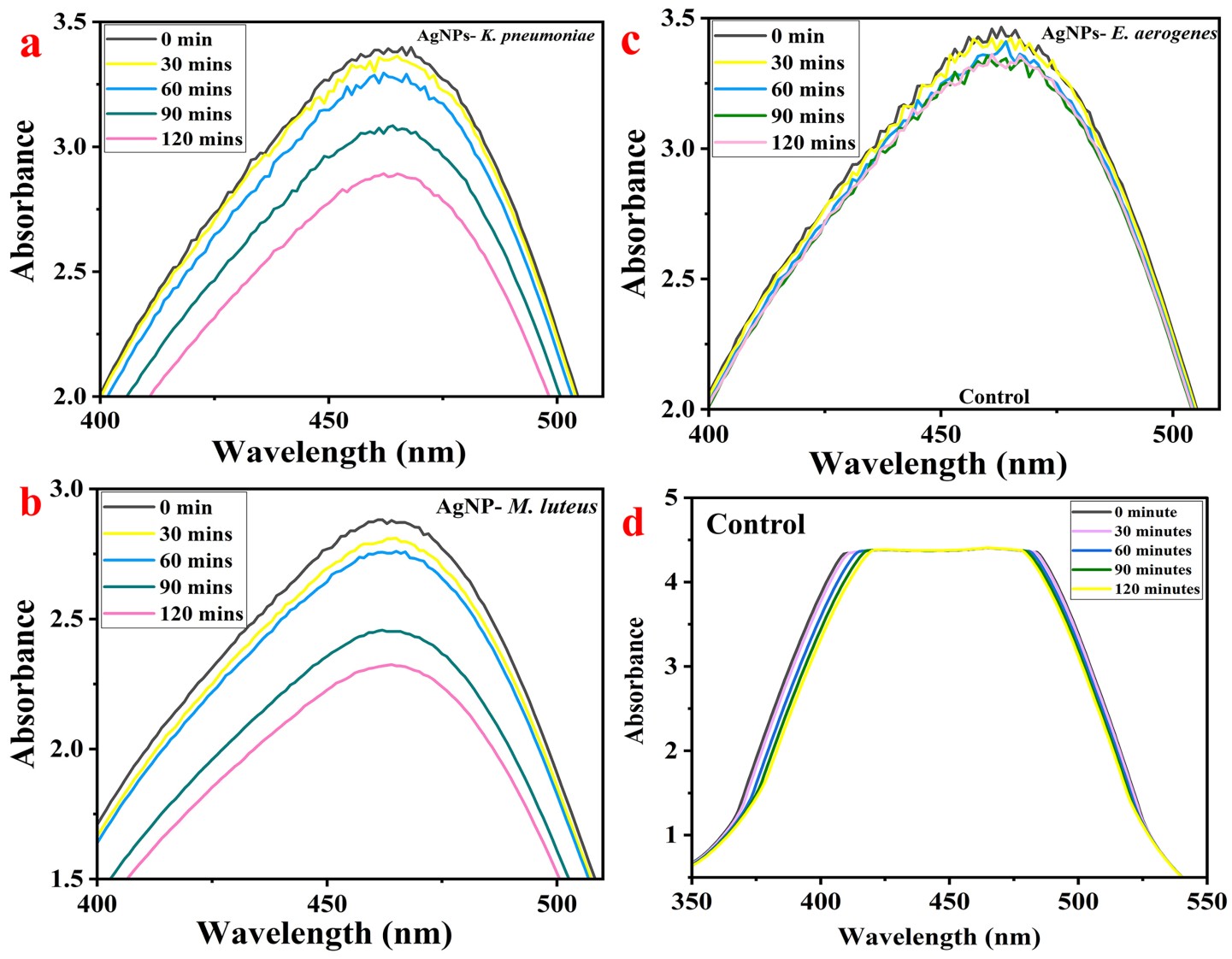

**Figure 6** MO dye removal by different types of AgNPs concerning contact time as measured by UV-Vis spectrophotometer: (A) AgNPs-K, (B) AgNPs-M, (C) AgNPs-E, and (D) control.

## Percentage removal of MR dye by all the AgNPs

AgNPs-M removed MO dye 2.34% at 30 min, 4.37% at 60 min, 14.83% at 90 min, and 19.24% at 120 min. AgNPs-K removed MO dye at 1.5% at 30 min, 4.06% at 60 min, 9.56% at 90 min, and 15.03% at 120 min. AgNPs-E removes MO dye 1.52% at 30 min, 2.36% at 60 min, 3.86% at 90 min, and 4.15% at 120 min. By comparing all the AgNPs mentioned above, it was observed that AgNPs-M has the highest dye removal efficiency and AgNPs-E has the lowest dye removal efficiency. Figures 7A–7C shows the percentage removal of MO dye by all three types of AgNPs.

AgNPs, synthesized by *Aspergillus* sp., photocatalytically degraded the reactive yellow dye in an aqueous solution where the initial dye concentration was about 20–100 mg/L, and about 1 g/L of retentate biomass of the fungus removed 82–100% dye. As the initial

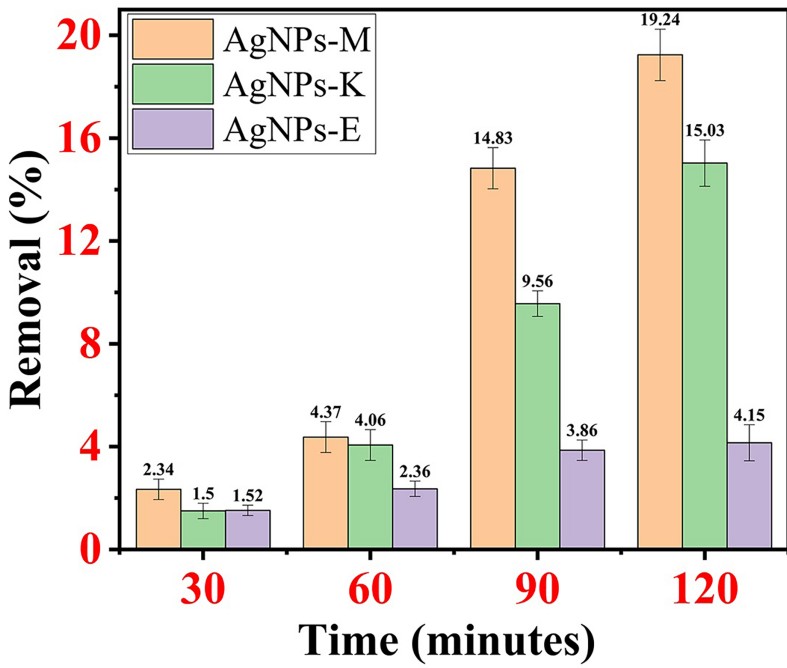

**Figure 7 Percentage removal of MO dye by all three types of AgNPs.**

dye concentration increased, the decolorization efficiency of the biomass retentate decreased from 9.2% to 32.3% (*Gola et al., 2021*). In one of the investigations, *Pseudomonas sp.* synthesized AgNPs achieved 100% removal of MB dye within 30 min (*Rasheed et al., 2023*), whose more detailed outcomes are provided in Table 4. AgNPs synthesized from *Salvinia molesta* demonstrated the maximum adsorption capacity (121.04 mg/g) of methylene blue dye on the surface of AgNPs by Langmuir isotherm (*Batool, Daoush & Hussain, 2022*). About 83% of MO dye was removed by using chemically synthesized AgNPs in the presence of $NaBH_4$ under optimized conditions (*Bhankhar et al., 2014*). Table 4 depicts the summarized form of all the investigations and current investigations where AgNPs were used for the removal of MO from wastewater.

Based on the investigations mentioned in Table 4, it was found that earlier, only a single attempt had been made to remove MO dyes from contaminated water by using AgNPs synthesized by chemical route, resulting in a removal percentage of 83% within a span of 2 min (*Bhankhar et al., 2014*). There was only one attempt where dyes (MB, 4-NP, RB5, and RR120) were removed by using AgNPs synthesized by *Pseudomonas sp.*, where the removal efficiency varied from 40–100% under optimized conditions (*Rasheed et al., 2023*). Moreover, in the current investigation, AgNPs-M has the smallest average size (21–45 nm), which exhibited the highest MO dye removal, *i.e.*, 19.24% in 2 h. In addition, the removal percentage of MO was just 4–19.24%, almost 5 to 15 folds less than the previous attempt by using AgNPs synthesized by the chemical method. The lower MO dye removal percentage in the current investigation might be attributed to the usage of a very low dose of AgNPs (1 mg/100 mL), in comparison to the previous study, whereas earlier, about 20 to

**Table 4 Summarized form of all the investigations and current investigations where AgNPs were used for the removal of various dyes from wastewater.**

| Synthesized method | AgNPs dose | Dye used | Initial dye concentration | Removal efficiency (%) | Contact time (minutes) | References |
|---|---|---|---|---|---|---|
| Chemical | | MO | 100 μL of $1 \times 10^{-3}$ M | 83% | 2 | *Bhankhar et al. (2014)* |
| *S. molesta* (plant) | 100 mg | MB | 10–100 mg/L | >96% (maximum at 100 mg/L) Adsorption capacity: 121.04 mg/g | 60 | *Batool, Daoush & Hussain (2022)* |
| *C. erectus* (plant) | 100 mg | Black 5 | 100 mg/L | 61.4% | 20 | *Rasheed et al. (2023)* |
| | 100 mg | RR 120 | 100 mg/L | 93.2% | | |
| | 20 mg AgNP + 1 mL NaHB$_4$ | MB | 50 mg/L | 49% | | |
| | 20 mg AgNP + 2 mL NaHB$_4$ | MB | | 70% | | |
| | 20 mg AgNP + 1 mL NaHB$_4$ | 4-NP | | 30% | | |
| | 20 mg AgNP + 2 mL NaHB$_4$ | 4-NP | | 42% | | |
| Chemical reduction method | 100 mg | Black 5 | 100 mg/L | 88.6% | | |
| | 100 mg | RR 120 | 100 mg/L | 53.7% | | |
| | 20 mg AgNPs + 1 mL NaHB$_4$g | MB | 50 mg/L | 35% | 20 | |
| | 20 mg AgNPs + 2 mL NaHB$_4$ | MB | | 65% | 20 | |
| | 20 mg AgNPs + 1 mL NaHB$_4$ | 4 NP | | 26% | | |
| | 20 mg AgNPs + 2 mL NaHB$_4$ | 4 NP | | 36% | | |
| *Pseudomonas sps.* (bacteria) | 100 mg | Black 5 | 100 mg/L | 88.6% | | |
| | 100 mg | RR 120 | 100 mg/L | 87.3% | | |
| | 20 mg AgNPs + 1 mL NaHB$_4$ | MB | 50 mg/L | 55% | 20 | |
| | 20 mg AgNPs + 2 mL NaHB$_4$ | MB | | 89% 100% | 20 30 | |
| | 20 mg AgNPs + 1 mL NaHB$_4$ | 4 NP | | 40% | | |
| | 20 mg AgNPs + 2 mL NaHB$_4$ | 4 NP | | 59% | | |
| Polyol reduction method | Avg. size: 70 nm + solar light | MB | | 94% | 80 | *Zaman et al. (2023)* |
| *K. pneumoniae* | 1 mg/100 mL | MO | 50 mg/L | 15.3% | 120 | Current investigation |

100 mg/50 mL of AgNPs synthesized from *Pseudomonas sp.* for the removal of varies organic dyes (20 mg AgNPs) (*Rasheed et al., 2023*). *Zaman et al. (2023)* and his research team successfully achieved a 94% degradation of MB in an aqueous solution using photocatalysis under solar irradiation.

To achieve a higher removal efficiency of dye, a higher dose of AgNPs is suggested. Moreover, further improvement in the dye removal by AgNPs could be done by using different types of chemical reducing agents (NaBH$_4$) along with the AgNPs (*Rasheed et al., 2023*). From the investigation, it was also concluded that the morphology of the AgNPs also affects the removal efficiency of the dyes, *i.e.*, smaller particles have a high removal efficiency (*Zaman et al., 2023*), as evident from the current investigation and study done by *Rasheed et al. (2023)*. Moreover, the removal of dyes under irradiation (solar, UV, *etc.*) may significantly enhance the dye degradation or removal, which could also be governed by pH and temperature.

## Evaluation of the antimicrobial activity of the synthesized AgNPs

AgNPs-K exhibits an impressive ZOI measuring 11 mm when tested against *B. megaterium* and *E. faecalis*. *B. subtilis* exhibits a moderate ZOI measuring 10 mm, while *B. cereus* demonstrates a 9 mm ZOI due to the impact of AgNPs. The AgNPs formulation (40–50 µg/mL) synthesized from *K. pneumoniae* exhibited higher antibacterial activity against *E. coli* in comparison to *S. aureus*, which might be due to variations in the thickness and composition of their cell walls, like peptidoglycan (*Sayyid & Zghair, 2021*). *Saleh & Khoman Alwan (2020)* used (50, 10, and 150 µg/mL) concentrations of AgNPs synthesized from *K. pneumoniae* and evaluated them against *E. coli, P. aeruginosa, S. aureus*, and *B. cereus* and found the highest concentration, *i.e.*, 150 µg/mL, was found to be most effective against these pathogens compared to 50 and 100 µg/mL. It can be inferred that a higher concentration of AgNPs leads to a greater level of antibacterial activity (*Saleh & Khoman Alwan, 2020*; *Liu et al., 2024*).

The utilization of AgNPs-M for studying antibacterial activity against various bacteria such as *B. subtilis, B. cereus, B. megaterium*, and *E. fecalis* reveals distinct variations in the size of ZOI. AgNPs-M shows a maximum ZOI of 12 mm against *B. megaterium* and a minimum ZOI against *B. cereus* of 8 mm. It offers a moderate ZOI of 9 mm against *E. faecalis* and 11 mm against *B. subtilis*. *B. megaterium* showed a higher susceptibility compared to *E. faecalis*, while *B. subtilis* displayed a moderate susceptibility, and *B. cereus* exhibited a lower susceptibility towards AgNPs-M. *B. cereus* gives a greater ZOI of 11 mm due to the effect of AgNPs-E. The smallest ZOI was observed while testing against *B. megaterium*, measuring 8 mm. Additionally, *B. subtilis* and *E. faecalis* both exhibit a ZOI of 10 mm.

Earlier *Aspergillus sp.* mediated synthesized AgNPs exhibited a ZOI of about 13 and 10 mm against *E. coli* and *S. aureus*, respectively, obtained for the *Gola et al. (2021)*. Additionally, the synergistic effect of AgNPs and penicillin against *E. coli* and *S. aureus* exhibited 0.49 and 0.36-fold increases in the ZOI, respectively, compared to the AgNPs and penicillin alone (*Gola et al., 2021*). *Paenarthrobacter nicotinovorans* MAHUQ-43 mediated synthesized AgNPs (13–27 nm, crystalline, spherical-shaped) showed antibacterial activity against *B. cereus* ATCC 10876 and *P. aeruginosa* ATCC 10145, where the MIC and MBC for both pathogens were 12.5 and 25 µg/mL, respectively. The ZOI at 30 µL against *P. aeruginosa* was 15.5 ± 0.8 and 13.6 ± 0.5 mm against *B. subtilis*, whereas at 60 µL the ZOI against *P. aeruginosa* was 24.7 ± 0.9 mm and against *B. subtilis* it was

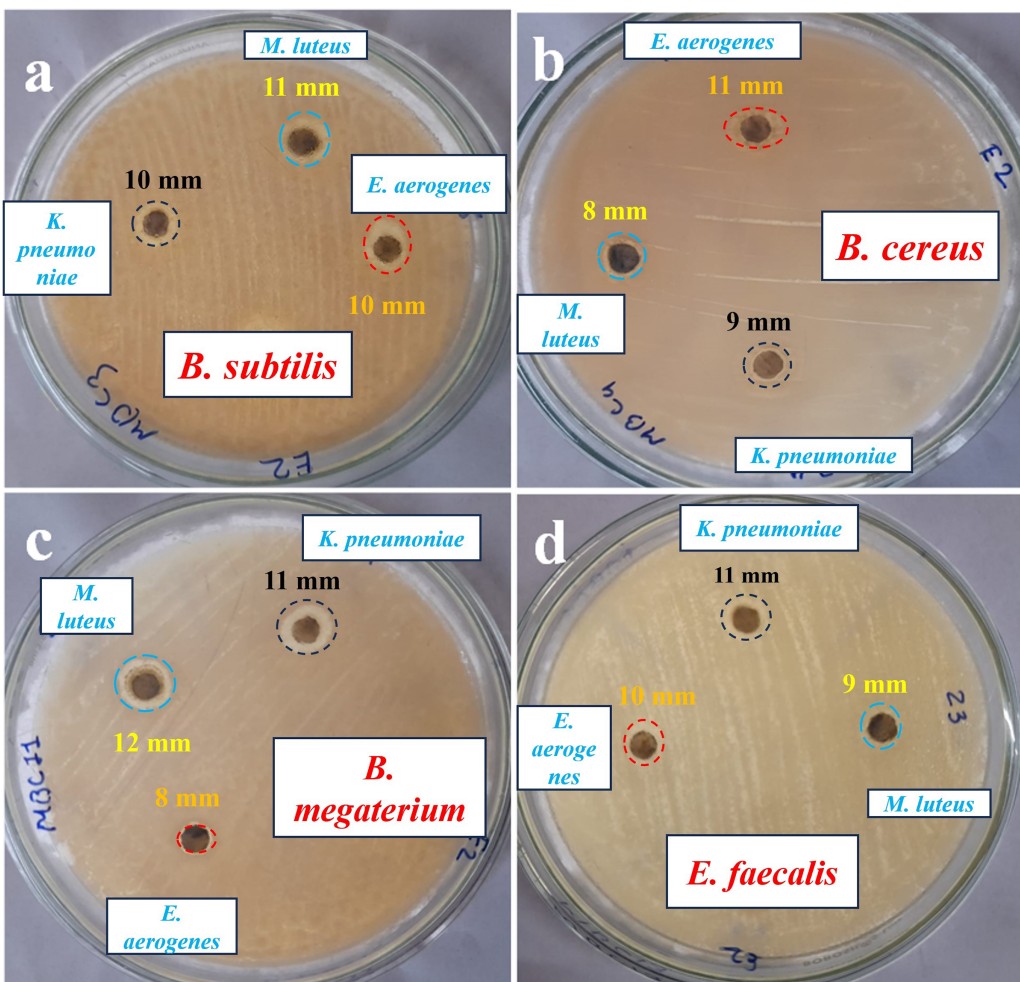

**Figure 8** **Antibacterial activity, zone of inhibition of AgNPs synthesized against the tested bacterial species.**

19.3 ± 1.0 mm (*Huq & Akter, 2021*). In one more investigation, AgNPs synthesized by *B. cereus* exhibited ZOI, *i.e.*, 32.12 ± 0.55 mm (*Staphylococcus epidermidis)*, 38.25 ± 0.05 mm (*S. aureus*), 33.05 ± 1.33 mm (*E. coli*), 30.73 ± 0.25 mm (*Salmonella enterica*), and 35.44 ± 1.08 mm (*Porteus mirabilis*) by agar-disc diffusion method at a concentration of about 200 µg/mL of AgNPs (*Ibrahim et al., 2021*). Figure 8 shows the antibacterial activity of Petri plates of the AgNPs against tested bacterial species. Table 5 shows a comparison of the antibacterial activity of the current investigation with previously reported studies.

The difference between the antibacterial effect against GPB and GNB bacteria over here might be due to the morphological differences in both types of bacteria (*Jubeh, Breijyeh & Karaman, 2020*). GPB has a thick peptidoglycan layer, while GNB has a thin peptidoglycan layer but a thick lipopolysaccharide layer (*Pasquina-Lemonche et al., 2020*). Several other investigators also provided a similar explanation for the antimicrobial activity of the synthesized AgNPs (*Kalpana & Lee, 2013*; *Singh & Mijakovic, 2022*). Investigators reported that spherical AgNPs have higher antibacterial/antimicrobial activity due to their

**Table 5 An overview of previous research comparing the antibacterial activity of AgNPs to the current study.**

| Tested organism | Synthesized by bacteria | Concentration of AgNPs (µg/mL) | Zone of inhibition (mm) | Method applied | References |
|---|---|---|---|---|---|
| E. coli, Ps. Aeruginosa, Staph. aureus & B. cereus | | 50 | Lowest | Agar-diffusion method | Saleh & Khoman Alwan (2020) |
| | | 100 | | | |
| | | 150 | Highest | | |
| Salmonella enterica, E. coli & S. pyogenes | | 50 & 100 | | | Kalpana & Lee (2013) |
| | | 40 to 50 µg/mL | | | |
| E. fergusonii | Bacillus cereus | MIC: 7.5 | | Tube dilution method | Gurunathan (2019) |
| S. mutans | | MIC: 9.5 | | | |
| E. coli | E. coli | | 10.0 | | Peiris et al. (2018) |
| Ps. aeruginosa | E. coli | | 10.0 | | |
| E. coli | Staph. aureus | | 14.7 | | |
| Ps. aeruginosa | | | 13.0 | | |
| Staph. aureus | | | 12.7 | | |
| MRSA | | | 12.7 | | |
| E. coli | Acinetobacter baumanii | | 13.3 | | |
| Ps. aeruginosa | | | 14.7 | | |
| E. coli | P. aeruginosa | | 13.0 | | |
| Ps. aeruginosa | | | 12.0 | | |
| Staph. aureus | | | 12.3 | | |
| MRSA | | | 12.7 | | |
| P. aeruginosa ATCC 10145 | P. nicotinovorans MAHUQ-43 | MIC: 12.5 µg/mL MBC: 25 µg/mL | 30 µL AgNPs: 15.5 ± 0.8, 60 µL AgNPs: 24.7 ± 0.9 | Disc diffusion | Huq & Akter (2021) |
| B. cereus ATCC 10876 | P. nicotinovorans MAHUQ-43 | MIC: 12.5 µg/mL MBC: 25 µg/mL | 30 µL AgNPs: 13.6 ± 0.5, 60 µL AgNPs: 19.3 ± 1.0 | Agar disc diffusion and broth dilution method | |
| Staphylococcus epidermidis | B. subtilis | MIC: 250 MLC: 400 | 32.12 ± 0.55 (200 µg/mL) | Agar disc diffusion and broth macro dilution method | Ibrahim et al. (2021) |
| S. aureus | | MIC: 200 MLC: 240 | 38.25 ± 0.05 (200 µg/mL) | | |
| E. coli | | MIC: 330 MLC: 520 | 33.05 ± 1.33 (200 µg/mL) | | |
| Salmonella enterica | | MIC: 220 MLC: 270 | 30.73 ± 0.25 (200 µg/mL) | | |
| Porteus mirabilis | | MIC: 140 MLC: 350 | 35.44 ± 1.08 (200 µg/mL) | | |
| B. subtilis | K. pneumoniae | | 10 | | |
| B. cereus | | | 9 | | |
| B. megaterium | | | 11 | | |
| E. fecalis | | | 11 | Agar-well diffusion | Current investigation |
| B. subtilis | M. luteus | | 11 | | |
| B. cereus | | | 8 | | |
| B. megaterium | | | 12 | | |
| E. fecalis | | | 9 | | |

| Tested organism | Synthesized by bacteria | Concentration of AgNPs (µg/mL) | Zone of inhibition (mm) | Method applied | References |
|---|---|---|---|---|---|
| B. subtilis | E. aerogenes | 10 | | | |
| B. cereus | | 11 | | | |
| B. megaterium | | 8 | | | |
| E. fecalis | | 10 | | | |

high SVR (*Al-Ogaidi & Rasheed, 2022*). Moreover, due to their large surface area, AgNPs could bind with various ligands. AgNPs can easily infiltrate the bacterial cell membrane, resulting in enhanced antimicrobial activity (*Kalwar & Shan, 2018*). This effect could potentially be enhanced even further by reducing the size of AgNPs (*Anees Ahmad et al., 2020*; *Wypij et al., 2021*). AgNPs get attached to the cell membrane, slowly releasing Ag ions, leading to the formation of pores in the cell wall of pathogens and ultimately enhanced permeability and leakage of cellular components through the plasma membrane, resulting in the death of the pathogens (*Tripathi et al., 2017*; *Chen et al., 2020*; *Matras et al., 2022*).

## CONCLUSIONS

The current investigation successfully exhibited that different bacterial spp., synthesize AgNPs with different morphology depending on the presence of biomolecules and enzymes in the bacteria. The synthesized AgNPs showed spherical, oval, and porous sheet-shaped type structures whose size varied from 10 nm to several microns. The bacterial synthesized AgNPs showed distinct UV-Vis absorbance maxima and well-defined crystal structures as revealed by XRD analysis. FTIR analysis confirmed the availability of various organic molecules with the synthesized AgNPs, suggesting their role in the formation and capping of AgNPs. The purity of the bacterially synthesized AgNPs varies (37.8% to 61.6 wt.%) depending on the organic molecule involvement, and washing during recovery of AgNPs as revealed by EDS. The AgNPs synthesized by *Micrococcus luteus* exhibited the highest percentage of MO dye removal, reaching up to 20%, and also exhibited maximum antimicrobial activity against *B. megaterium*, measuring 12 mm. The higher dye removal efficiency and antimicrobial activity of AgNPs-M could be due to the smallest size of the AgNPs produced by *M. luteus*. Therefore, the morphology of the AgNPs significantly influences the dye removal percentage efficiency and antimicrobial activity. Such bacterial-based green synthesis of AgNPs acts as a sustainable approach for material synthesis with a potential in the field of biomedicine.

### Funding

This research was funded by the Deanship of Scientific Research at King Khalid University under the grant number R.G.P. 2/174/44. The funders had no role in study design, data collection and analysis, decision to publish, or preparation of the manuscript.

### Grant Disclosures

The following grant information was disclosed by the authors:
Deanship of Scientific Research at King Khalid University: R.G.P. 2/174/44.

### Competing Interests

The authors declare that they have no competing interests.

### Author Contributions

- Bhakti Patel conceived and designed the experiments, performed the experiments, prepared figures and/or tables, and approved the final draft.
- Virendra Kumar Yadav conceived and designed the experiments, performed the experiments, prepared figures and/or tables, and approved the final draft.
- Reema Desai performed the experiments, authored or reviewed drafts of the article, and approved the final draft.
- Shreya Patel performed the experiments, authored or reviewed drafts of the article, and approved the final draft.
- Abdelfattah Amari analyzed the data, prepared figures and/or tables, authored or reviewed drafts of the article, and approved the final draft.
- Nisha Choudhary performed the experiments, authored or reviewed drafts of the article, and approved the final draft.
- Haitham Osman analyzed the data, prepared figures and/or tables, authored or reviewed drafts of the article, and approved the final draft.
- Rajat Patel performed the experiments, authored or reviewed drafts of the article, and approved the final draft.
- Deepak Balram analyzed the data, prepared figures and/or tables, and approved the final draft.
- Kuang-Yow Lian analyzed the data, prepared figures and/or tables, and approved the final draft.
- Dipak Kumar Sahoo conceived and designed the experiments, analyzed the data, prepared figures and/or tables, and approved the final draft.
- Ashish Patel conceived and designed the experiments, analyzed the data, prepared figures and/or tables, and approved the final draft.

### Data Availability

The raw data are available in the Supplemental File.

## Supplemental Information

Supplemental information for this article can be found online at http://dx.doi.org/10.7717/peerj.17328#supplemental-information.

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
