# Peer review of "Bacteriogenic synthesis of morphologically diverse silver nanoparticles and their assessment for methyl orange dye removal and antimicrobial activity"

_PeerJ, doi:10.7717/peerj.17328_

## Round 0.1 · original submission · Major Revisions

Dear Authors,

Thank you for submitting your work to PeerJ. Your study has undergone review, and the feedback received recommends a 'Major revision.'

Please find the constructive comments from the reviewers 1,2 and 3, mainly suggesting that the manuscript needs proofreading for English. Here are a few key points that I found particularly interesting: Please address the issue of novelty raised by reviewer 2. Reviewer 3 questioned the main claim of this study. Additionally, technical information, such as 'bacterial isolation and characterization,' was highlighted for revision by reviewer 1.

I kindly request that you respond to each of the comments of reviewers 1,2, and 3. I am looking forward to seeing your revision.

Best wishes,
Dr. Nagendran Tharmalingam
Editor, PeerJ

**Language Note:** The Academic Editor has identified that the English language must be improved. PeerJ can provide language editing services - please contact us at [email protected] for pricing (be sure to provide your manuscript number and title). Alternatively, you should make your own arrangements to improve the language quality and provide details in your response letter. – PeerJ Staff

·

Basic reporting

I appreciate the authors for their extensive work and sharing their work along with the raw data. However, the text lacks proper usage of grammar and sentences formation at many places. This has been annotated throughout the attached PDF. The scope of improvement is beyond my annotated comments. The authors also need to address the comments below before this manuscript could be accepted for publication.

Experimental design

The study fails to explain how does it fills up an existing research gap or adds value to the present knowledge and benefit the community. Rather as the references suggests, similar work has already been reported by other groups.

Validity of the findings

Specific comments related to the validity are as follows:
1. The methods did not mention how does the authors characterize the three bacteria they obtained from soil and what made them to pursue it further.
2. Section 2.2.6 (line 190) The disks were prepared but finally the liquid was used. What for the disks used, need to write the text better so as to avoid confusion
3. Line 261 contradicts the figure as the peak of AgNPs-E is at 503 nm
4. The authors mention that ratio of AgNO3 and bacterial supernatant affects absorption intensity (line 271) but they did not explain why do they get a peak for AgNPs-E at 503 nm. Is it because of this reason? Have they checked this?
5. Section 4.6 (line 422) is difficult to comprehend
6. Figure.9 lacks labelling and hence, difficult to make sense.
7. The conclusion need to highlight how the research findings fills up an existing gap or describe the novelty of the study with its benefit to the community.

Additional comments

Please revise as per concerns above. Take help from annotations made throughout the PDF attached hereby.

·

Basic reporting

In this manuscript, Patel et al. has utilized soil-isolated gram-positive and gram-negative bacteria to synthesize silver nanoparticles and studied their efficiency in dye removal. It is commendable that authors have utilized various techniques for characterization of the synthesized nanoparticles.

Major concerns:
Questions – Usage of silver nanoparticles to remove synthetic dye is an interesting aspect of the study. It would be more interesting to discuss literature related to environment friendly nature of silver nanoparticles and their removal thereof.
The English language should be improved (eg. lines 386, 479, 486, 377, 364, 316, 289, 251, 147, 148, 138, 127 – 129, 122 – 123, 104, 90, 91 etc). I found the manuscript to be wordy. Coherence and continuity of the text requires improvements.

Repeated sentences – 387 and 367.

Experimental design

I appreciate authors for including enough literature knowledge. How is this study different from existing studies should also be included. Information or data such as presence of any reductase enzyme in bacterial supernatant is lacking, in addition to enzymes such as Lactamase, Lipase, Protease etc.

Validity of the findings

Although authors have provided a table (Table 3) of information containing literature study, for the interest of readers, it would be interesting to describe the correlation or contradictions of the findings with existing literature.

Additional comments

Minor concerns:
I suggest to use universal reference format (line 185, 105 etc.) in the main text of the manuscript as well as in the reference section (eg lines – 510, 512, 553, 556 etc).

Authors have described the results of published work, however there is no explanation about how it is related to the current study.
Please improve incomplete sentences eg. Line 393.

Name of all bacteria should be mentioned in a conventional format for eg. P. aeruginosa (italics), rather than in user defined acronyms.
No explanation on AgNPs-K and AgNPs-M in the abstract was established.

Reviewer 3 ·

Basic reporting

The manuscript displays a certain degree of redundancy and repetition. Additionally, there is a lack of consistency in the nomenclature of microorganisms, and some abbreviations are not adequately defined. References to studies by other researchers are made without appropriate citations.

1. Short, one sentence description of the other methods (chemical and physical) for AgNP production with references
2. Comparison of these two methods with biological method and the advantage of biological method over chemical and physical methods.
3. There are two whole paragraphs in introduction that summarize the antimicrobial properties of the AgNPs however not much is discussed in the text regarding the use of AgNPs for dye removal. Authors should consider reducing the two paragraphs to one and only include the most unique and important studies in context of the manuscript. The text contains some redundancy in terms of antimicrobial properties of AgNPs. This concept is repeated multiple times.
4. I would like to recommend expanding upon the existing information and referencing studies that discuss the use of AgNPs in dye removal. Currently, the manuscript might give the impression that this is the first study exploring the use of AgNPs for dye removal. However, it's important to acknowledge that there is a substantial body of research on this topic, which can provide valuable context and insights. A simple online search reveals several studies that have successfully employed AgNPs for dye removal, and incorporating these references would enhance the completeness of the manuscript.

Experimental design

5. In 2.2.1, were the colonies grown on the agar plate after serial dilution? What was the composition of the media? How were the colonies isolated and identified? Please include this information.
6. 2.2.2 may not be needed and can be included in 2.2.1 or removed altogether.
7. Line 156 mentions three bacterial supernatants, these are not mentioned anywhere in the method text before this. What are the three supernatants? Three different microorganisms? Same microorganism but different culture? What was the incubation time?

Validity of the findings

8. Line 185 provide proper reference with the publication year.
9. Section 2.2.6, It would be helpful to have a more detailed explanation of the experimental setup. Specifically, it would be valuable to know whether there were two distinct methods used – one involving dried AgNPs on discs and another involving liquid AgNPs. Additionally, information about the inoculum size used for plating is missing. Were the bacteria pre-cultured in liquid media overnight and subsequently diluted to meet certain McFarland Standards before plating on the agar? These details would enhance the comprehensibility of the methodology.
10. Line 262 add reference for Saleh and Alwan.
11. Section 4.2, all the sentences (Line 262-268)that mention the wavelength of AgNPs by citing various studies could be consolidated into one sentence and added all the references to that for brevity and avoid redundancy.
12. The sentences in line 428 to 432 are repetitive. Consolidating all these sentences into one sentence would be better.
13. Adding the size of the ZOI in figure 9 would make it more informative to the readers.
14. The conclusion should provide a more complete overview. Currently, it summarizes the results but doesn't explain why this study is important or how it adds to existing knowledge. It would be helpful to discuss the future implications of the research and the authors' main goals.

Additional comments

The manuscript needs major revision.

---

## Round 0.2 · Major Revisions

Dear Authors,

I would like to bring to your attention that the submitted revision has not addressed the potential comments made by reviewer 1. Kindly take the necessary actions to address these comments.

Additionally, it is suggested to provide a concise summary for both Figure 1 and Figure 2, as neither figure appears to present substantial data.

It has been observed that Figure 9 and Table 4 present identical results. Please either eliminate one of them if this observation is accurate or provide a justification.

Tables 1, 3, and 5 may be incorporated into the discussion section, as they currently do not present any experimental data.

Following the team's peer review, we have determined that the manuscript is still in the stage of major revision. We request that you carefully address the mentioned issues and make the necessary revisions. We look forward to receiving a manuscript of high scientific quality.

Best regards,
Dr. Nagendran Tharmalingam
Handling Editor, PeerJ.

·

Basic reporting

The language still needs a lot of improvement which is evident by the comments in the attached PDF. Overall, the writing approach is very casual and does not seem to fit for publication in a journal of international repute.

Experimental design

The authors still fails to justify the novelty of the study and how the study fills up an existing research gap.

Validity of the findings

The findings are difficult to comprehend more because of the improper use of English language. The study doesn't seem to be novel as suggested by the references used and conclusions drawn.

Additional comments

Unless, the study fails to justify it's novelty and improvement in writing, it does not seem fit for publication in the present state.

·

Basic reporting

I appreciate authors effort on improving the manuscript.

Experimental design

Experimental design and data analysis were performed according to the standards.

Validity of the findings

Authors have made effort in synthesizing silver nanoparticles using bacteria for the purpose of bioremediation of synthetic dyes, it is indeed an interesting subject for the readers.

---

## Round 0.3 · Minor Revisions

Dear Authors,

I hope this message finds you well. We have received the reports from peer reviewers, and it appears that the manuscript still requires significant improvements, particularly in terms of language. We appreciate your efforts and dedication to this work and encourage you to approach the revisions meticulously.

To facilitate a smoother process, we kindly request that you focus on enhancing the language aspects, ensuring clarity, coherence, and precision in your writing. Additionally, please address any specific feedback provided by the peer reviewers to strengthen the overall quality of the manuscript.

Your commitment to refining these elements will undoubtedly contribute to the success of your work. If you have any questions or require further guidance during the revision process, feel free to reach out. We believe in the potential of your research and look forward to seeing the enhanced version of your manuscript.

Best wishes for your continued efforts and success in refining your work.
Dr. Nagendran Tharmalingam
Handling Editor.

·

Basic reporting

While the authors have been able to justify the gaps in the field, the use of English language still needs improvement. As for an example, where the authors cite other people, they write, 'Esmail and their team....' while it should be, 'Esmail and his team....'. This mistake including many others were pointed out in the annotated PDF previously but has not been corrected since then. The approach in writing should be more formal than casual for scientific publications. The authors may take help from professional/native speakers for improvement in writing for plausible publication of their work in PeerJ.

Experimental design

The conclusion is ending abruptly.

Validity of the findings

No comment

Reviewer 3 ·

Basic reporting

The authors have put generous effort into improving the introduction section and details of methodology.

Experimental design

Methodology is more elaborate than previous version and seems to be easier to understand.

Validity of the findings

The manuscript has substantially improved. The use of nanoparticles for dye removal and as antimicrobials is interesting avenue to explore.

---

## Round 0.4 · Minor Revisions

Dear Authors,

Please have a look at reviewer 1 comments and address the critical points.

Kindly,
Handling Editor.

·

Basic reporting

Please check the comments in the attached manuscript for further improvement of the language

Experimental design

No comment

Validity of the findings

No comment

---

## Round 0.5 · Minor Revisions

Dear Authors,

After two rounds of review solely focusing on language, it is evident that the paper requires a more thorough and concentrated examination. Kindly refer to the comments provided in Review 1 and articulate your responses to acknowledge the efforts of Reviewer 1

Kindly,
Dr. Nagendran Tharmalingam
Handling Editor.

**Language Note:** The review process has identified that the English language must be improved. PeerJ can provide language editing services - please contact us at [email protected] for pricing (be sure to provide your manuscript number and title). Alternatively, you should make your own arrangements to improve the language quality and provide details in your response letter. – PeerJ Staff

·

Basic reporting

Dear Authors,

I am sorry to write this but after so many of rounds of revision, the manuscript is expected to be well proof-read and free from typos and grammatical errors. As suggested earlier also, either the manuscript should be dealt by professionals or an expert having a different pair of eyes than the one who is always addressing these concern. As evident below, the manuscript still has many errors which shows the casual approach in handling/addressing the comments.
Figures:
Figure 1: 3rd line (orange)....the word 'three' typed as 'there'
Figure 4: Figures should be self explanatory which is not the case here. Please describe the major characteristics bands in this figure legend.
Figure 5: Highlight characteristic peaks and differences in figure legend.
Figure 8: Lacks error bars or was the dye removal experiment done only once?
Text:
Line 75-78- Incomplete sentence
Line 90: utilize of....?
Line140-143: Incomplete sentences starting from 'Moreover,....
Line 151: GPB and GNB should be elaborated for the first time
Line 152: Gola and their.....Gola is just one person!
Line 163: (EDS). AgNPs. ....?
Line 180: were procured or were obtained??
Line 205: Nothing has been mentioned about control while it was mentioned to be used
Section 4.2: The authors missed describing their own findings and describe highlights of Figure 3 while only stating others work
Line 377: Lacks reference by Vimalanathan.
Line 502: p value
Line 543, 544: Difficult to comprehend
Line 633: needs reframing
It is advised that the authors should review everything beyond the above comments before resubmitting the manuscript for publication while utilizing the submission window maximally. Hoping to see a better and the final version of the manuscript this time.
All the best

Experimental design

No comments

Validity of the findings

No comments

---

## Round 0.6 · Minor Revisions

Dear Authors,

Please have a look at review 1's comments and respond carefully.

Best wishes.
Dr. Nagendran Tharmalingam
Academic Editor.

·

Basic reporting

Dear Authors,

I do not find the updated figures attached. The manuscript still has the old figures as opposed to the revised ones mentioned in the rebuttal letter.

Moreover, when we talk of pronouns and mentioning something like Gola and their team..., we are mentioning Gola as more than one person because the pronoun, 'their' is used for Gola here and not his team. I do understand that Gola has worked with 'his' team as you mentioned in the rebuttal letter. Hope this helps!

Experimental design

NA

Validity of the findings

NA

Additional comments

Please share updated figures

---

## Round 0.7 · Minor Revisions

Dear Authors,

Thank you for submitting your revised version. After careful consideration of appeal, we found the below comments were unnoticed in section 1. Please find the comments from Editorial board/ reviewer board in section 2 requests the critical changes to pursue further.

Section 1.
1. Review Round 1 9/29/2023
a. The study fails to explain how does it fills up an existing research gap or adds value to the present knowledge and benefit the community. Rather as the references suggests, similar work has already been reported by other groups.

The comment need to be addressed- "how does it fills up an existing research gap or adds value to the present knowledge"

b. 7. The conclusion need to highlight how the research findings fills up an existing gap or describe the novelty of the study with its benefit to the community.

The comment need to be addressed- "describe the novelty of the study with its benefit to the community"

2. Review Round 2 12/6/2023
a. Unless, the study fails to justify it's novelty and improvement in writing, it does not seem fit for publication in the present state

The comment need to be addressed- "the study fails to justify it's novelty"

Section 2.
Additional comments.

3. Figure 1 shows the methodology that has already been mentioned in the Methods section. However, it does not display any potential results that support the study. Therefore, it should be deleted

4. Figure 2 show the "development of AgNPs from silver ions by bacteria via the NADH-dependent nitrate reductase enzyme".

Line 314-315: Please provide experimental evidence for the "development of AgNPs via bacterial NADH-dependent nitrate reductase enzyme" conducted for this study. Did the authors isolate the "bacterial NADH-dependent nitrate reductase enzyme" and perform in vitro enzymatic assays to confirm its activity? If the authors failed to provide this evidence, it would be better to delete the figure2.

5. The introduction spans around 6 pages, with nearly 1640 words, resembling a thesis. Please trim the introduction to fit a manuscript format

6. The selection of bacteria is not clearly outlined, as mentioned in the Methods section, "Around 10 bacterial colonies were procured on nutrient agar Petri plates, which were stored in a refrigerator in the laboratory". The genus, species, was not mentioned.

6a. The phrase lacks proper grammar. "How can colonies be procured? Bacteria can be purchased either in lyophilized form, as glycerol stocks, on agar slants, or on agar plates.

7. Please clarify the genus and species of bacterial strains in "2.2.2. Synthesis of silver nanoparticles from bacteria. For the fabrication of AgNPs, silver salt was reduced by the bacterial supernatants obtained from all three bacterial strains"

The author mentioned "K. pneumoniae, M. luteus, E. aerogenes, B. subtilis, B. cereus, B. megaterium, and Enterococcus fecalis" in the Materials section 2.1. Please clarify what bacteria was used in section 2.2.2. The genus name and species name should appear in full-form when mentioned for the first time and can be abbreviated thereafter. Please adhere to the proper usage of genus and species

8. Line 251-256- "Firstly, 16 discs of a specific size (8 mm diameter) were cut out of filter paper and dipped into separate reagent vials containing AgNPs-K, AgNPs-M, and AgNPs-E. Further, all the reagent vials were sonicated for 15-20 minutes using an ultrasonicator (Lequitron). Further Additionally, the AgNPs loaded discs were taken out of the vials with the help of forceps kept on three different Petri plates and dried in a hot air oven at 40-50 ℃"

Did the authors know the concentration of relevant agents impregnated in the disc? Or is it just a random concentration? Please clarify the exact concentration of AgNP (microgram/mL or mg/mL) that inhibited bacterial growth. Alternatively, please indicate whether the term "random concentration" or "relevant word usage" is appropriate.

9.Figure 8 depicts the dye removal comparison among three treatments. However, I do not observe the control group (with no AgNP), to compare against the AgNP-treated group

10. The entire manuscript is excessively lengthy and requires trimming. Please limit the corroborative stories to their results. The study included numerous corroborative stories with results, contributing to the manuscript's excessive word count

· Appeal

Appeal

Dear Dr. Uversky and Dr. Tharmalingam,

We express our apologies for the confusion and missing one of the reviewer's comments. We missed a typo in Figure 1 (which was uploaded separately, not with the main manuscript file). We thoroughly reviewed each one, addressed each comment, and made the necessary revisions to the manuscript and other files. Kindly refer to the files attached to this email. We would highly appreciate it if you could kindly reconsider this submission.

Looking forward to your kind response,

Many thanks,

Kind regards,

Dipak


· · Academic Editor

Reject

Dear Authors,

Thank you for your diligent efforts in revising the manuscript in response to the feedback provided. We truly appreciate the time and dedication you've invested in addressing the concerns raised by both the editorial team and the reviewers. While we acknowledge the sincerity with which you approached the revisions, it appears that there are still some significant issues present within the manuscript that have not been fully addressed.

The editorial team and reviewers have highlighted several areas where improvements were needed, and unfortunately, it seems that the responses and revisions provided may not have fully met the level of rigor expected. We understand that navigating the intricacies of scientific discourse can be challenging, and we commend your efforts to sharpen and refine the scientific content. However, it's important to ensure that not only the science itself but also its presentation and responsiveness to critiques are robust and thorough. In light of the current state of the manuscript, regrettably, the editorial team and reviewer team are unable to pursue further consideration for publication at this time.

We understand that this decision may be disappointing, and we sincerely hope that you will not be discouraged from submitting future work to our publication. Your contributions to the field are valued, and we encourage you to continue refining and strengthening your research. Once again, we thank you for your submission and the opportunity to consider your work. Please do not hesitate to reach out if you have any questions or if there are any further clarifications needed.

Warm regards,
Dr. Nagendran Tharmalingam
Handling Editor.

·

Basic reporting

Figure 1: 3rd line (orange)....the word 'three' typed as 'there'

Persisting since the first draft

Experimental design

NA

Validity of the findings

NA

Additional comments

NA

---

## Round 0.8 · accepted · Accept

Dear Authors,

Thank you for submitting the manuscript with the addressed issues. We are pleased to inform you that your work has been accepted for publication with PeerJ. The production team will be in contact with you shortly regarding typesetting queries.

A kind suggestion for future reference: when responding to reviewers, please include the page number and line number in the rebuttal letter where the authors made edits. This will assist the review team in locating changes efficiently, minimizing time spent on finding specific locations.

We look forward to seeing your future work with PeerJ.

Best wishes,
Dr. Nagendran Tharmlalingam,
Handling Editor